# Impacts of the Establishment of Biofoulants on Greek Aquaculture: Farmers' Expert Knowledge

Dimitrios Tsotsios [1], Dimitrios K. Moutopoulos [1,*], Athanasios Lattos [2], Basile Michaelidis [2] and John A. Theodorou [1]

1   Department of Fisheries and Aquaculture, University of Patras, 30200 Mesolongi, Greece; dtsotsios@yahoo.gr (D.T.); jtheo@upatras.gr (J.A.T.)
2   Laboratory of Animal Physiology, Department of Zoology, School of Biology, Aristotle University of Thessaloniki, 54124 Thessaloniki, Greece; aqualattos@gmail.com (A.L.); michaeli@bio.auth.gr (B.M.)
*   Correspondence: dmoutopo@upatras.gr

**Abstract:** Ascidians' bioaccumulation is frequently responsible for the massive growth of certain species, causing detrimental effects on aquaculture facilities. The goal of this study is to provide, for the first time in the Eastern Mediterranean, information on biofoulant species in Greek mussel farms over a long time scale and to describe the best management strategies that will reduce costs while preventing and controlling these biofoulants. An interview survey was conducted to assess mussel farmers' expert judgment on non-endemic ascidians as well as their opinions on the magnitude of the invasion's impacts. The results show that ascidians and, to a lesser extent, sponges exhibited the highest intensities in mussel farm units during the last 20 years, whereas gastropod invasion was limited and observed after 2015. Ascidians exhibited the most significant impact on the final product, whereas sponges showed a moderately negative impact, with reduced amounts of flesh being the most important effect. The cost of farming management only rose with ascidians and sponges and was mostly impacted by damages to maintenance and labor and, to a lesser extent, fuel. All invasive species affected the operational cost of production at a rate of 21–50%, which peaked from July to September. The above problems are increasingly aggravating in cases where farm units undergo production shutdown due to plankton bloom. Preventive management action against the establishment of biofoulants in Greek mussel aquaculture is of paramount importance.

**Keywords:** invasive species; ascidians; NIS; Rapana; sponges biofouling; mussel farming; CE Mediterranean; aquaculture; mussel farming

## 1. Introduction

### 1.1. The Problem

Open marine aquaculture systems offer limited control over cultured organisms and relatively few mechanisms to regulate biological environment interactions [1]. Reduced productivity as a result of biological interactions with invasive organisms, including biofoulants, is an ongoing challenge, which is exacerbated by the intensification of cultivation [2–4]. Ascidians are among the most important biofoulants in aquaculture and also serve as artificial marine reefs [5] for these species groups [6]. Ascidians are also known for their high spatial activity [7], and such artificial activities enhance the spread of invasive ascidians [8–10], constituting one of the primary routes of non-indigenous species introduction [2]. Ascidians' bioaccumulation is frequently responsible for the massive growth of certain species, causing detrimental effects on aquaculture facilities (e.g., floating platforms and aquaculture piers), with both economic and ecological consequences [11–15]. To mitigate these effects on aquaculture, a series of measures are implemented that may raise the cost of production to the final price of the product by 5–10% for fish farms and 20–30% for shellfish [6].

Ascidians are important survivors in mussel farms, with significant economic implications for the produced product, while these facilities are also thought to be important substrates for the establishment and spread of these species (Figure 1). Some ascidians, such as *Didemnum vexillum* (Venice lagoons [16]; Ebro Delta [17]; *Styela clava* (NW Mediterranean [18]; and Sea of Marmara [19]) and the "old invasive but also cryptogenic" *Clavelina* spp. in the Mediterranean [20], population out-breaks of which covered farmed mussels [21,22], lack knowledge regarding their biology and genetics [14,15,20]. It is worth noting that *Didemnum* spp. ascidians are among the species that, according to the Horizon scanning exercise, we expect to spread in European seas [23–25] and that, as biofoulants, cause serious damage to mussel farms [26]. Because of their widespread distribution, the aforementioned ascidians are potentially invasive species in the Eastern Mediterranean as well. As a result, the development of primarily molecular methods for species identification is required so that mussel farms can be validly identified. Valid warning (the early warning system) is used as a principle in mussel farming [27], with the goal of rapid environmental risk assessment, and it is an important tool for the immediate adoption of unit management measures [28], and it has also been adopted as an EU strategy [29,30].

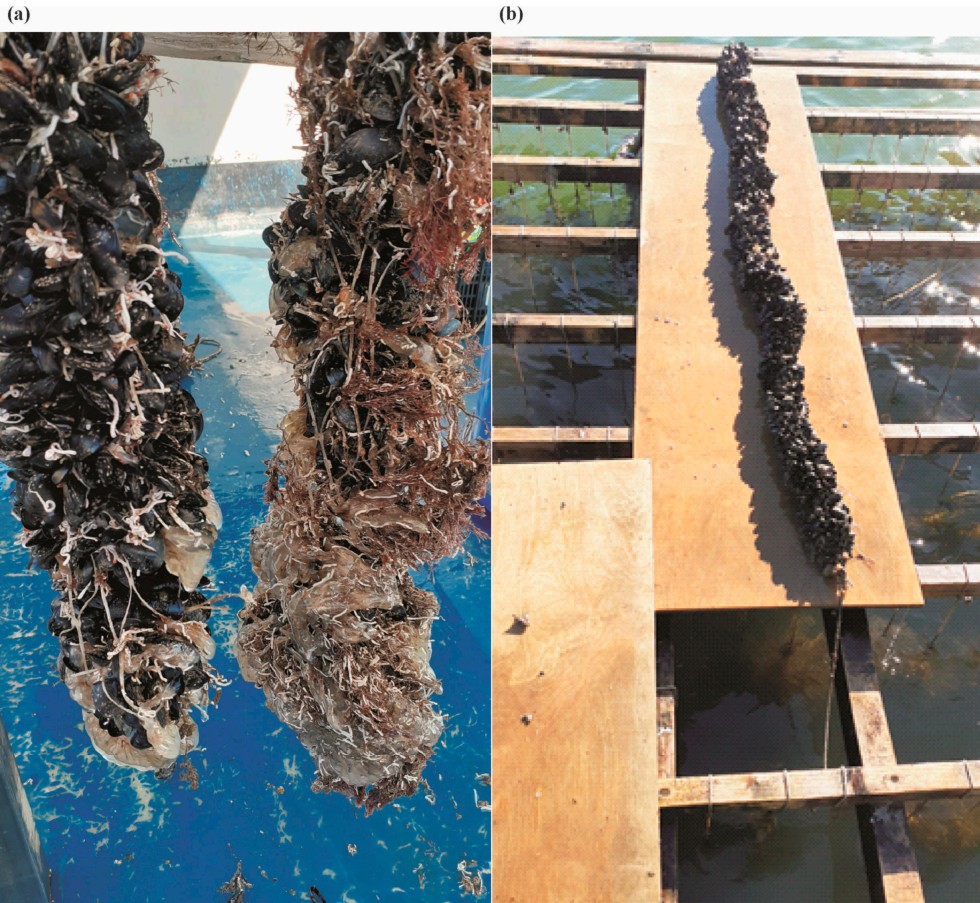

**Figure 1.** Biofouling treatments on mussel socks in Amvrakikos Gulf (see Figure 2, region no. 7) mussel farm: (**a**) left image shows treatment handling (pressure water washing and air exposure) once a month during mid-summer (July–August 2022), and right image shows untreated control at the same time; (**b**) treated whole mussel sock in early November with additional treatment handling in September.

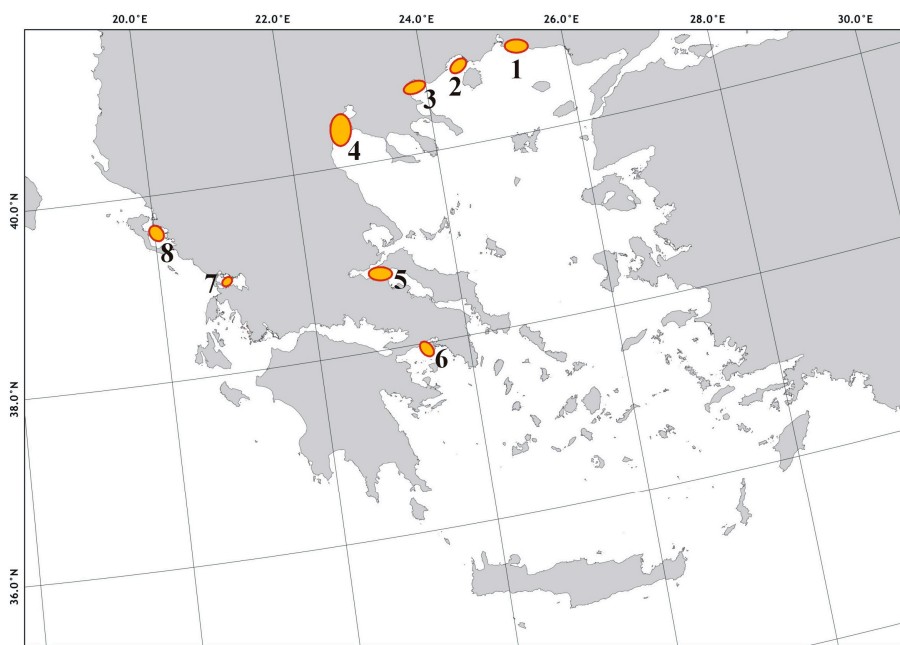

**Figure 2.** Sampling points of shellfish farms participating in the survey (grey circles): Vistonikos (1), Kavala (2), Strymonikos (3), Thermaikos (4), Maliakos-Evvoikos (5), Saronikos (6), and Amvrakikos (7) Gulfs, as well as Sayada Strip in Thesprotia (8).

### 1.2. Focus Area

In the Greek Seas, 75 species of ascidians have been recorded [31], and the composition of the species mainly concerns invasive species (44.4% are of Atlantic–Mediterranean origin) while 40.3% are indigenous [32]. This ratio has shifted in favor of invasives as new records of species of Indian origin (via Suez) emerge [16,33]. Other information is limited and focuses on faunal lists or publications with a primary focus on ecology [32,34–37], binary populations [38–40], reproduction [38,39], or interactions with other invertebrates [41,42]. Population outbreaks during their hosting in mussel farm facilities cause a slew of management issues during the breeding and sorting of farmed products [23,43,44]. Mediterranean mussel aquaculture in Greece is mostly established in seven bays (Thermaikos, Maliakos, Saronikos, Amvrakikos, Strymonikos, Kavala, and Vistonikos: Figure 2) and annually produces from 16,000 to 23,000 t of mussels valued between EUR 6 to EUR 8.5 million (45) with a total production capacity of approximately 40,000 tons [45,46]. These systems receive freshwater inputs that are rich in nutrients from river outflows, creating an environment that is suitable for high productivity [46]. To support traditional shellfishery harvesting and the relatively newly established mussel/oyster farming, allocated production zones have been established in these regions. Continuous monitoring of the environmental, health, and safety procedures according to EU legal standards qualify the suitability of the products for commercial exploitation for human consumption [47–49]. The main shellfish species of the North Aegean based on their commercial value in combination with their abundance are the mussel (*Mytilus galloprovincialis*), the flat oyster (*Ostrea edulis*), the warty venus (*Venus verrucosa*), the horse mussel (*Modiolus barbatus*), the callista (*Callista chione*), the well-known Noah's Ark (*Arca noae*), the smooth scallop (*Chlamys glabra*), the bean or tellina (*Donax trunculus*), etc. However, despite being designated as protected species under P.D. A92/29-04-02, the date mussel (*Lithophaga lithophaga*) and the fan mussel pinna (*Pinna nobilis*) are nonetheless illegally harvested because of market demand. Both species are well-known both in the domestic and export markets [50–52].

These farms serve as substrates for the establishment of a variety of biofoulants, including ascidians [6], causing functional issues in production. Together with ascidians, the sponge *Paraleucilla magna*, which is sometimes mistaken for an ascidian, and the

gastropod *Rapana venosa*, which have been investigated for their potential spread and effects on Greek mussel farms, have also been studied [53]. Given that the farmed organism itself serves as a substrate for biofoulants to settle in mussel farms, the treatment includes a variety of exposure actions for the mass of farmed species, such as exposure to air, pressure washing, and chemical use. These actions are related to either the different resistance of the survivors and farmed species [23] or reducing the possibility of settlement (e.g., changing the rearing depth in areas where the abundance of the survivors' pelagic forms is low) [6].

### 1.3. Aim of the Study

The goal of this study is to provide, for the first time in the Eastern Mediterranean, information on biofoulant species in Greek mussel farms over a long time scale and to describe the best management strategies that will reduce costs while preventing, controlling, or eliminating populations of biofoulants to avoid having a negative material economic impact on the mussel farms' outputs.

Actions to mitigate the effects of biofoulants are integrated into unit management. Thus, the development of integrated approaches to fishing exploitation mechanisms that take into account the local empirical knowledge of professionals has become increasingly popular (fishers' ecological knowledge [54]). A contemporary method for gathering data on fisheries is to quantify the expertise of shellfish producers (shellfishers and farmers), and this approach is being integrated more and more into fisheries and ecological research [54,55]. The contribution of mussel farmers is necessary because they can provide valuable information on changes in the composition of biofoulant species in their facilities and tools, as well as on production fluctuations over time. Given that farmers serve as direct users of biological resources when producing food from farming, the FAO recognizes them as important contributors to sustainable development through their interactions with scientists and other interested parties [56]. These methods aid in disentangling the effects of invasive species from those caused by environmental factors, paving the way for the development of strategic management techniques to maximize the long-term performance of farmed bivalve stocks.

The indirect effects of biofoulants may extend far beyond shellfish farming because their proliferation on farms may lead to subsequent infestations of adjacent natural ecosystems [57]. Thus, information such as that which the current study provides could also be incorporated into ecosystem management models (regarding Greek waters, see [58]), as it can create reference limits and indicators for exploitation to comprehend the structures and functions of marine ecosystems (e.g., [59]).

## 2. Materials and Methods

### 2.1. Survey Design

A semi-structured questionnaire was used (Supplementary Materials) to assess mussel farmers' expert judgment on non-endemic ascidians as well as their opinions on the magnitude of the invasion's impacts [60,61]. The collection of marine professionals' experiences with the spread of non-endemic ascidian species and its impact on them and their activities were included in this study. Questionnaires utilizing the technique of oral and/or telephone interviews were given to shellfish farmers from the following areas: the Vistonikos, Saronikos, Strymonikos, Thermaikos, Kavala, Maliakos-Evvoikos, and Amvrakikos Gulfs, as well as in Sayada Strip in Thesprotia (Figure 2).

The respondents that participated in the survey and were located in the aforementioned areas that were chosen represented at least 30% of the total allowed capacity or 30% of the total number of shellfish farming units [46]. Individual shellfish farmers and representatives of associations and cooperatives took part in this study by completing a single questionnaire that reflected all these participants' opinions. To determine the sample from the target groups to complete the questionnaires, official updated data regarding the shellfish farming units in the target areas were recorded following the official data from

the Fisheries Departments of the Regional Agriculture and Veterinary Directorates of the respective regions, as well as the updated contact details of the relevant collective bodies.

The survey used the two most popular data collection methods: in-person interviews and postal delivery of questionnaires (including via email). The first method is more dependable but more expensive, whereas the second method is less expensive but has lower reliability and response rates. Another survey approach that was used was telephone interviews, which combine the unreliability of a direct interview with the affordability of emailing. It should be noted that most of the required data were gathered through direct interviews.

The project team then conducted 28 interviews with mussel farm owners, covering the entire Greek coastline (Figure 2). In particular, the sample collected from the Thermaikos Gulf represents up to 30% of all licensed shellfish farming units and roughly 68% of the total allowed yearly capacity [46]. The sample taken from the Strymonikos Gulf corresponds to 22% of the total annual production capacity and 33% of the licensed operating shellfish farming units. The Amvrakikos Gulf sample represents 52% of the total annual production capacity and 50% of the active unlicensed shellfish farming units. The sample taken from the Vistonikos and Kavala Gulfs (numbers 1 and 2 in Figure 2) represents 100% of the total number of licensed operating shellfish farming units as well as 100% of the total annual production capacity. The sample collected from the Sayada Strip and the Kalama Estuary of Thesprotia is equal to 29% of the total number of licensed operating shellfish units and slightly more than 30% of the total approved annual capacity. For the Saronikos Gulf, the collected sample represents 25% of the total number of licensed operating shellfish farming units and 33% of the total allowed annual capacity. The sample taken from the Maliakos-Evvoikos Gulfs (number 5 in Figure 2) represents more than 30% of the total licensed active shellfish farming units and more than 30% of the total allowed yearly capacity.

*2.2. Interview Survey*

Interviews were carried out from February to June 2022, and before completing the questionnaires, the mussel farmers were informed that participation in the survey was voluntary and that the survey was impersonal. To minimize any potential bias, all interviews were carried out by the same person, ensuring that questions were presented identically and freely answered with no prompt or influence.

The questionnaire consisted of three sections. Six questions grouped under "farming specifics of each unit" were included in the first section. They pertained to the farmed species, production capacity, annual production, unit area, installation depth, area of activity, unit start-up, farming system method, and farming equipment. The second section of the questionnaire contained 10 questions on "on-growing issues" such as "Which species of ascidians/gastropods/sponges are identified as a problem in the rearing process" (ranked from 1 = most significant to 3), "Has their abundance changed in the last five years?", "Are invasive species present every year (in the last five years)?", "What is the impact of the presence of invasive ascidians/gastropods/sponges on the production process?", "What is the impact of the presence of invasive ascidians/gastropods/sponges on the production process?", and "Since when is the problem significant for each of the above organisms?".

The third section of the questionnaire contained nine questions about the final product and the economic return to reverse the problem including the following: "Evaluate the magnitude of the following species' detrimental effects on the final product", "What factor and to what extent does the presence of invasive ascidians/gastropods/sponges effects?", "For which of the species and by what percentage does the cost of on-going management increase?", "Estimate the percentage (%) burden on the operational costs and sales from the impact of the above organisms", "Propose an alternative management scenario for the impact of invasive species on the production process", and "What would be your pro-invasive species management strategy?".

*2.3. Data Analysis*

The frequency of preferences in the whole sample was estimated at the level of the independent variables mentioned above, and for their presentation, the method of correlation tables was chosen, as well as that of diagrammatic presentation. At the same time, an independence test was performed with the $\chi^2$ distribution (likelihood ratio $\chi^2$) for each of the preference questions and with the independent ranking variables of the respondents [62]. In cases of statistical significance, an analysis of the "Adjusted Standardized Residuals" was performed in comparison to the observed frequencies related to consumer preferences. A non-parametric correlation analysis (Spearman's rank correlation coefficient ρ) on the values of the questions on the self-efficacy scale was also carried out to determine whether any correlations existed and to assess the strength of the relationships [62].

A reliability analysis of the multi-thematic questions was also conducted. This analysis refers to the property of a measurement that causes it to give similar results for similar inputs. Cronbach's alpha coefficient is a measure of reliability, which is defined as the proportion of variability in the responses to a survey, which is the result of differences in the respondents [63].

Multi-topic questions were those that included more than one topic to assess different levels of statements. These questions were "What is the effect of the presence of invasive ascidians/gastropods/sponges on the production process?" and "Which factor and to what extent does the presence of invasive ascidians/gastropods/sponges affect?". These questions were answered on a three-point scale (low, moderate, or high).

Multi-variate analyses were also applied to the multi-thematic questions. The categorical regression method with optimal scaling constitutes an improvement and extension of the classic linear regression method, which quantifies the data on categorical variables by attributing numerical values to the categories, resulting in an optimal linear regression equation of the converted variables. This method also allows forecasts of the values of a dependent variable for any combination of a set of independent (classification) variables to be made [51,64]. The effect of each of the classification variables on the dependent variable is described with the corresponding regression coefficient. To test the collinearity in the model, Pratt's measures of relative importance and tolerance were used. A variable with a very low tolerance contributes little information to a model and can cause computational problems. Thus, it should be removed from the categorical regression.

All of the analyses were carried out using the statistical package IBM SPSS Statistics 27.0.1.0 [65].

**3. Results**

*3.1. Typology of the Shellfish Farming Units*

A great part of the interviewed shellfish farm owners (92.9%) focused on mussel farms, and a smaller percentage (7.1%) also included oyster farms. The capacities of the breeding units that took part in this study ranged from 25 t to 12000 tons, with most of them (50.0%) exhibiting capacities between 100 and 150 tons. The yearly output for 2020 was in line with the past year, ranging from 25 t to 115 t, with most of the farms (58.0%) producing between 100 and 150 t. The surface areas of the breeding units that took part in this study ranged from 1 to 170 hectares (henceforth, ha), with most of the farms (57%) covering an area of 2.0–2.5 ha. The depths at which the units were installed ranged from 3 to 25 m, with most of the units (29%) being installed at 15 m to 25 m.

The spatial distribution of the units covered 7 areas (Figure 2): the Thermaikos Gulf (40.8%), Strymonikos Gulf (13.6%), Gulf of Ambracian (4.5%), Kavala-Keramoti and Vistonikos Gulfs (22.7%), Sayada Strip and Kalamas Estuary (9.0%), Saronikos Gulf (4.5%), and Maliakos-Evvoikos Gulf (6.9%). The first mussel units were established in 1988 (25.0%), while most of them (45.0%) were established in the 2000s. Almost all of the farming units (94.0%) were floating, and only one employed the longline method.

### 3.2. Problems in Farming

In most units that participated in this research, the predominant difficulty (96.0%) consisted of ascidians and a considerably smaller number of gastropods (4.0%). In the breeding units, gastropods (12.6%) and sponges (87.5%) were regarded as the 2nd and 3rd major issues, respectively. The 3rd issue in the breeding units was identified as gastropods (65.0%), with sponges (20.0%) and ascidians (15%) coming in 2nd and 3rd place, respectively.

According to the owners of the shellfish farming units, the issue of the establishment of invasive species in the mussel farms dates back to 2000. Most of the participants indicated that the problems derived from ascidians began in the years 2010–2013 (55.6%), 2013 (40%), and 2012 (50.0%) for gastropods and sponges, respectively (Figure 3a). The commencement of operation and the date of the invasive species problem in the shellfish farming units were found to be significantly ($p > 0.05$) correlated in the respondents' responses. The size of the shellfish farming unit was significantly ($p > 0.05$) and negatively correlated with the timing of the introduction of invasive ascidians (Spearman's ρ = −0.575) and sponges (Spearman's ρ = −0.487), both of which first appeared in larger shellfish farms earlier in time. The inverse was true for gastropods (Spearman's ρ = 0.603), which appeared in newly established farm units.

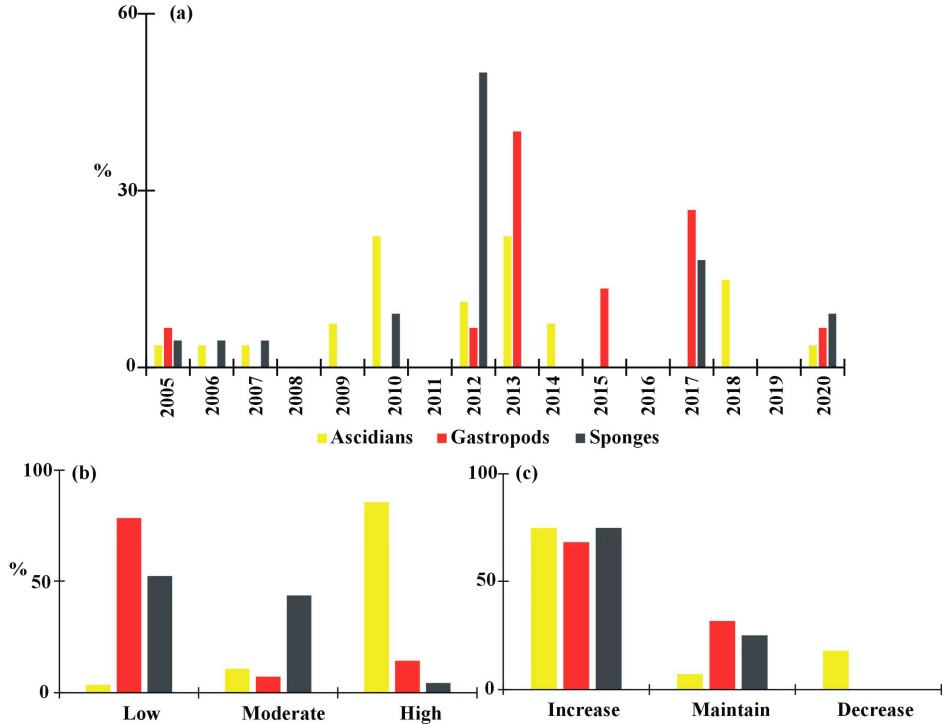

**Figure 3.** (**a**) Chronological occurrence of invasive species presence in mussel farms, (**b**) degree of intensity of the problem for each of the invasive species, and (**c**) percentage change in the invasive species' abundance during the last 5 years.

Most of the participants (85.7%) rated the problem's severity as severe in the case of ascidians, but it was moderate for sponges (43.5%) and low for gastropods (78.6%) (Figure 3b). Participants noticed a rise in the abundance of all three studied biofoulants when asked about the change in their abundance over the previous five years (Figure 2c). This fact was also corroborated by the consensus of an overwhelming number of the owners of the shellfish farming units (94.7%) that the invasive species were introduced in the last 5 years.

Grading the intensity of the problem (1: low, 2: moderate, or 3: high) of the invading ascidians or sponges in relation to the size of the unit (Figure 4a), the problem was significantly ($\chi^2$, $p < 0.05$) higher in units with areas of 2.0 and 2.5 ha (64.6% of the total), while it

was moderate in units larger than 5.0 ha (Figure 3a). For sponges, on the other hand, the problem was significantly ($\chi^2$, $p < 0.05$) higher only in the largest farms, while the intensity of the problem was moderate in the smallest (under 2.5 ha) (Figure 4b). In terms of the impact of invasive species on monthly production (Figure 4), the impact peaked from July to September (>57.1%), followed by October and November (32.1% and 53.6%, respectively) (Figure 5).

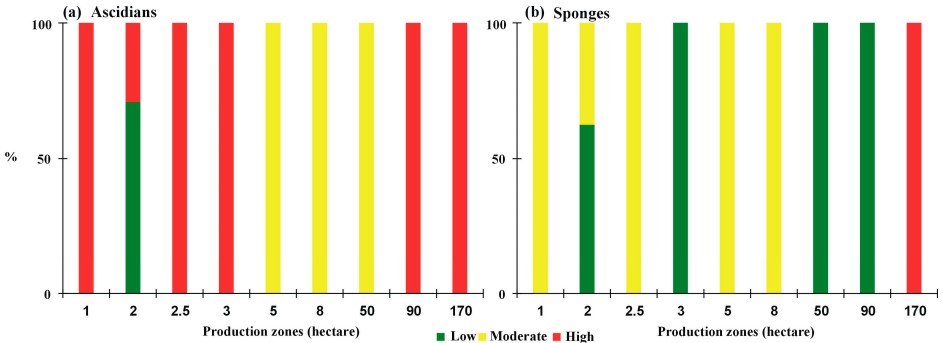

**Figure 4.** Quantifying the intensity of the problem (1: low, 2: moderate, or 3: high) of the invading ascidians or sponges in relation to the size of the shellfish farm unit.

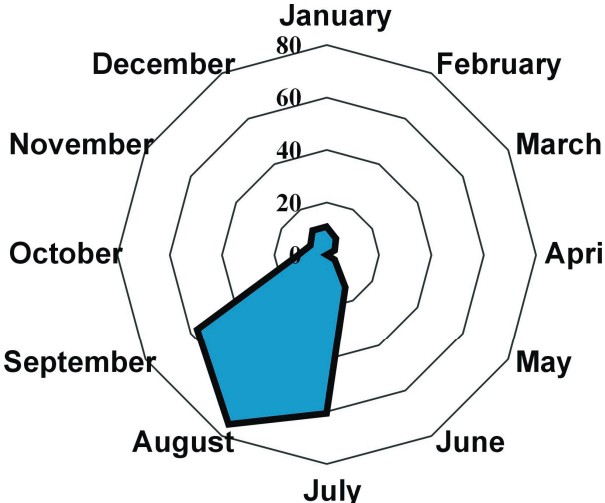

**Figure 5.** Monthly impact (%) of invasive organisms on shellfish farms' production.

The participants' personal opinions on managing the influence of invasive species on the production process in mussel farms suggested obtaining detailed knowledge of the invasive species, cleaning the farmed organisms, as well as acquiring equipment such as extra vessels. They also advised linking the management of the units with climatic phenomena, as some participants linked the increase in the development of invasive species with the increase in temperature. The participants also recommended support from local stakeholders in the fight against invasive species, collaboration with scientific organizations, the immediate organization of Areas of Organized Aquaculture Development (AOAD), and investment in cutting-edge cleaning products.

### 3.3. Final Product and Cost to Address the Problem

A high percentage of the owners of the units stated that ascidians exhibited the highest degree of negative effect on the final product (71.4%), followed by sponges (66.6%), with a moderate effect, whereas gastropods did not exhibit any sign of effect (95.5%) (Figure 6a). Likewise, most of the participants (>60%) stated that only ascidians and sponges increase the cost of farming management (Figure 6b), with ascidians costing more (30–50%) and

sponges costing less (11–30%). In contrast, for gastropods, all participants stated that the cost of farming management does not increase significantly (0–10%) (Figure 6b). Most of the participants (50%) claimed that all invasive species affect the operational cost of production at a rate of 21–50% (Figure 6c). This statement relates to the extent to which the presence of invasive species has an impact on this cost.

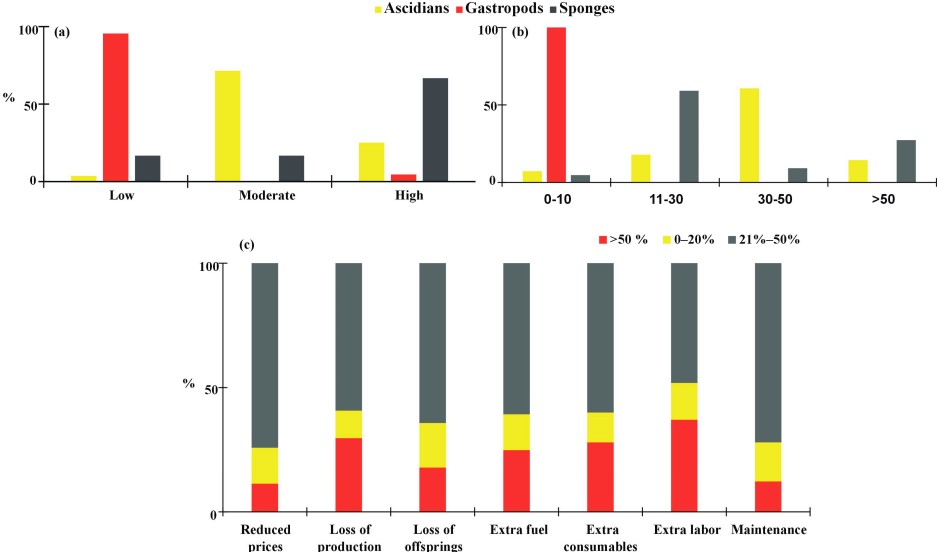

**Figure 6.** Grading the impact of negative effects on shellfish farms (**a**) for each invasive species on (**b**) the percentage increase in management cost for each invasive species and (**c**) the operational cost of each part of the production chain.

Investigating the impacts observed from the presence of invasive species on the production process for ascidians, the greatest part of the owners stated that the biggest issue is the overlap of the farmed species (92.9%), with the growth rate reduction (46.4%) and the increase in the farmed species mortality (32.1%) coming in 2nd and 3rd place, respectively (Figure 7a). The impact on the production process for gastropods was deemed to be moderate, particularly concerning the overlap (60.9%) and rising mortality (60.8%) of the farmed species (Figure 7b). For sponges, 25% of the participants stated that the effect was highly significant for the overlap of the farmed species, while more than half (57.1%) stated that it was moderate for the farmed species' rising mortality (Figure 7c). Most of the participants chose washing (>55.67%), frequent grading (>50.0% as a 2nd choice), and brushing (>48.0% as a 3rd choice) as the best ways to deal with invasive species during production (Figure 6d–f). As an alternative strategy, the owners of the units stated that more treatment procedures were being used, including manual removal (>57.1%) and sun exposure (>56.0%).

Most of the participants who responded to the question about the factors that are significantly impacted by the presence of invasive organisms stated that in the case of ascidians, the effect was large for the reduced amount of flesh (28.6%) (Figure 8a); for gastropods, the effect was large for the reduced product shelf life (66.7%) (Figure 8b); and for sponges, the effect was large for the variation in farmed sizes (26.1%) (Figure 8c). Ascidians, however, exhibited a moderately negative impact on the reduced flesh quantity (57.1%) and unattractive look (57.1%) (Figure 8a). Likewise, most of the participants (>60%) reported that only ascidians and sponges exhibited an increase in the cost of farming management, with ascidians (Figure 8d) depicting a greater rise (30–50%) and sponges (Figure 8e) a lower increase (11–30%). In contrast, all participants stated that the expense of managing the farms did not greatly rise (0–10%) for gastropods (Figure 8f). Most participants (>50%) indicated that ascidians (Figure 8d) and sponges (Figure 8e) exhibited a moderate impact on maintenance, labor, and fuel expenses, while gastropods (Figure 8f)

showed a substantial impact on packing and sales when it comes to the degree to which the presence of invasive species affects annual production costs.

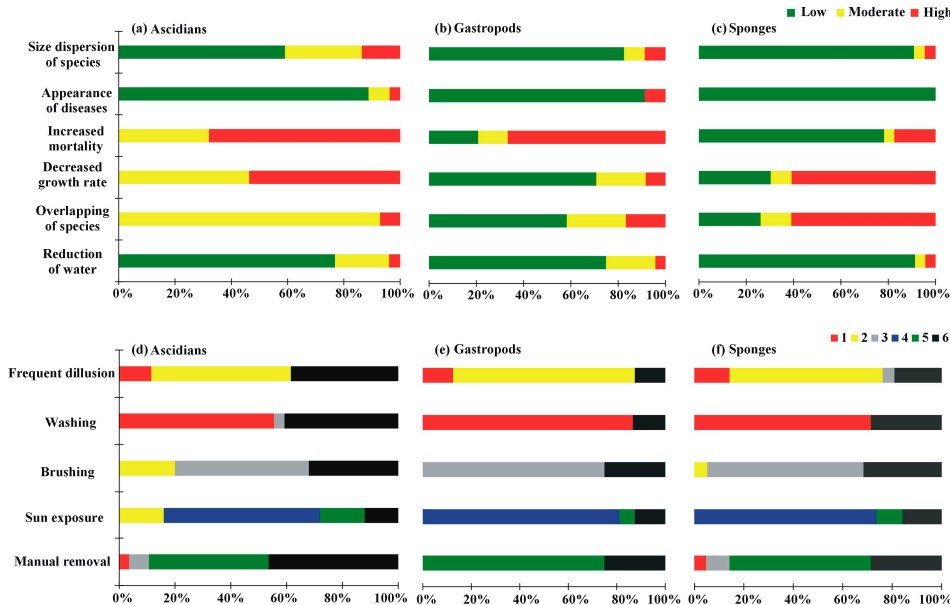

**Figure 7.** Grading the negative impact observed from the presence of invasive species on the production process (**a**–**c**) and the best practices for dealing with the management of invasive species (**d**–**f**) (1: first choice, . . . 6: last choice).

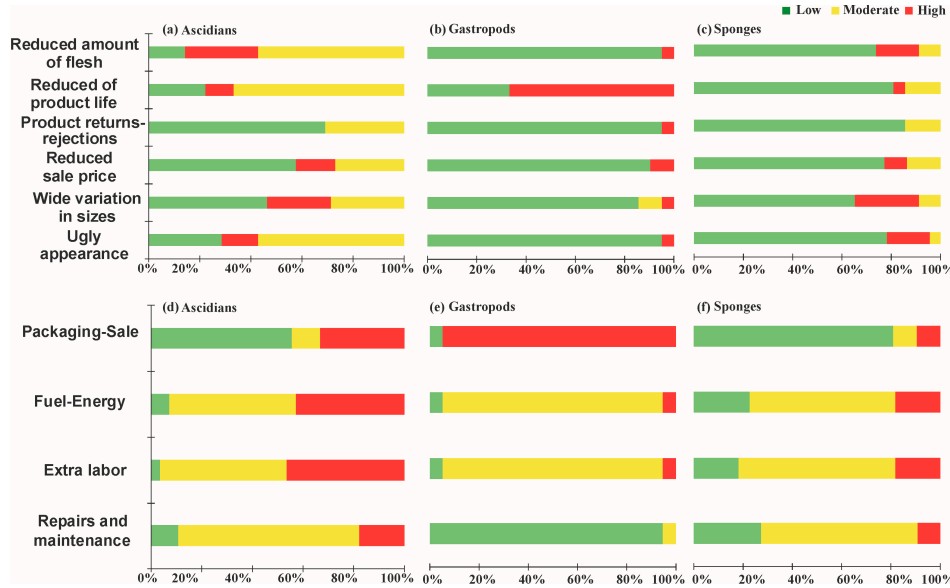

**Figure 8.** Grading the negative impact observed from the presence of invasive species on the quality of the final shellfish product (**a**–**c**) and on the annual cost of each chain of shellfish production (**d**–**f**).

## 4. Discussion

The present study's goal was to provide, for the first time in the Eastern Mediterranean, information on biofoulants species in Greek mussel farms and to describe the best management strategies that will reduce the costs of the end-product while controlling the biofoulants' populations. The information provided by the professionals consists of one of the most direct approaches to activating the links between mussel farmers and scientists to upgrade the quality of the information and data provided, especially in data-poor areas such as those of shellfish farms [50].

The principal problem in the Greek shellfish farms was ascidians and, to a lesser extent, sponges. The surface size covered by a shellfish farming unit was also significantly and negatively correlated with the time of appearance of invasive ascidians, where ascidians appeared earlier in the larger units. The problem was more intense in farm units covering an area between 2.0 and 2.5 ha, whereas it seemed moderate in units over 5.0 ha. This might be explained by the fact that the smaller-sized farms have built-in routine technologies (e.g., regular cleaning, etc.) as mitigation measures for the establishment of biofoulants. Ascidians also exhibited the highest intensity during the last 20 years, whereas gastropods were the invasive species with the highest intensity in the units that first began operation more recently (after 2015). In terms of the final product, ascidians exhibited the most significant impact, whereas sponges showed a moderately negative impact, with the effect of the reduced amount of flesh being the most important.

The inherent challenges in shellfish farms, where management interventions are costly and difficult to implement, the rapid invasive parasite recolonization, and the potential for off-target effects pose regulatory barriers to many mitigation measures [64]. Worldwide, in most developing countries, invasive species groups pose immediate threats to productivity and profitability through competition for food or space, predation, product degradation, and interference with economic turnover [6,66,67]. Our results show that the cost of farming management rises for ascidians and sponges and is mostly impacted by damages to maintenance and labor and, to a lesser extent, fuel. The studied invasive species affected the operational cost of production at a rate of 21–50%, peaking from July to September, whereas the previous (June) or following (October) months were identified as the 2nd most significant temporal influence on production. The greatest impact of the presence of invasive species was related to repairs—maintenance, labor, fuel, and service—during the harvest season. Extra labor from May to June due to invasive ascidians added extreme threats to the profitability of shellfish farms with additional economic costs. According to [68], who collected data from eight farms and modeled the costs according to the sizes of the shellfish units, labor costs represented a fourth of the total farm costs in an average farm of two hectares. In the same context, [66] surveyed commercial bivalve farms in the United States, and more than 90% of the respondents reported operational problems caused by invasive species, with biowaste management accounting for approximately 15% of their production costs. Because of the unsightly appearance of the end product and buyer rejections, invasive ascidians reduce marketability [69]. The cost-to-benefit ratio was 1:6.3, indicating a 16% chance of success in eradication [70].

The above problems are becoming more aggravating in the cases where a unit undergoes production shutdown due to plankton bloom [45,47], with periodic production shutdowns remaining the primary tool for agencies tasked with public protection in cases where HABs may affect human health. The impact of harmful algal blooms (HABs) on mussel farmers is generally related to their impact on profitability due to the periodic cessation of cultivation [71]. In Greek coastal waters, 16 algal species have been identified as responsible for HABs [72], and in the mussel farming area of Chalastra (Northern Greece [47]), profit losses in scenarios corresponding to yearly harvest bans of 45 to 165 days (6 to 22 weeks) varied from 4% to 38%. Sale limitations lasting for a period of 6 weeks might be detrimental in the period from early spring (March) to summer (July–August). In this context, even a shorter closure period (4–6 weeks) applied during harvest season could result in comparable outcomes [54]. Given that the most important periods in a shellfish farm are between May and August [54] and the impact of invasive organisms is primarily concentrated between July and September, a special management plan is required, which could be extended until October, as this month has been designated as the second most important temporal impact on production [23]. At the same time, preventive management action for early-developing mussels could be implemented during the May–June period, as the month preceding the above period (June) is considered important in terms of the effect of invasive organisms, primarily ascidians.

The findings of the current study also coincide with those of [73], who concluded that the frequency and duration of economic losses from mussel production in Galicia (Spain) as a result of harvest prohibitions connected to HABs varied. Regarding the "sensitivity" of each season, risk evaluations of production halts cause spatial restrictions on long-line farms during this time, and if harvest delays are mandated, there is no room available for fresh seed stock. In the case where a farm is not harvested, the seeds that are still in the collectors grow more quickly in warm climatic conditions and finally sink to the bottom of the farm. When all producers scramble to sell their goods as quickly as possible after the farm restarts, further losses are anticipated owing to declining pricing. When mussels do not take the time that they should, further losses occur, particularly in the summer when heat waves and damage to mussel socks "pergolari" (tubed mussels in cylindrical plastic nets) due to strong winds in late July and early August (Etesian winds) cause a rise in mortality. A temporal shift during harvest could be used to handle brief incidents, decreasing losses. When the shutdown period in Greece exceeds 4–6 weeks, summer is the season most vulnerable to catastrophic losses.

To conclude, management interventions can result in significant net economic benefits. Monitoring *M. galloprovincialis* production could provide an empirical quantitative estimate of the economic impact of invasive species [74]. Although there are many variations of integrated invasive organism management, the basic principles are combined risk assessment and risk management activities based on five main pillars: a thorough understanding of the invasive organism ecology, bioeconomic cost–benefit relationships, continuous monitoring at the right scale, active prevention, and reactive control. These principles could be applied to shellfish farming, with experts' judgments and knowledge playing a critical role in long-term approaches to invasive organism management. When it comes to understanding the ecology of invasive organisms, the effects of both bio-pollution and invasive organisms tend to vary spatially, temporally, and/or demographically. Thus, site-to-site variation factors, such as species' plasticity to adapt to different environmental conditions, could also help to predict and assess the risk in farmed areas exposed to invasive organisms [75–78]. By establishing predictive relationships between the entry of an invasive organism, its yield, and economic impacts, bioeconomic cost–benefit relationships between production impacts could be created.

**Supplementary Materials:** The following supporting information can be downloaded at: https://www.mdpi.com/article/10.3390/jmse11051077/s1.

**Author Contributions:** Conceptualization, D.T., J.A.T. and D.K.M.; methodology, D.T. and D.K.M.; validation, D.K.M., J.A.T. and B.M.; formal analysis, D.T. and D.K.M.; investigation, D.T. and A.L.; resources, D.T. and A.L.; data curation, D.T., D.K.M. and A.L.; writing—original draft preparation, D.T. and D.K.M.; writing—review and editing, J.A.T., D.K.M. and B.M.; visualization, D.T. and D.K.M.; supervision, J.A.T., D.K.M. and B.M.; project administration, J.A.T.; funding acquisition, J.A.T. All authors have read and agreed to the published version of the manuscript.

**Funding:** The present work is a part of the project «Development of the best control practices of invasive ascidians in mussel farming infrastructures and remediation of economic effects of invasion» (code MIS: 5048463) funded by the EU–Greece Operational Program of Fisheries, EPAL 2014–2020.

**Institutional Review Board Statement:** Not applicable.

**Informed Consent Statement:** Not applicable.

**Data Availability Statement:** The data supporting the reported results of this study can be provided upon request to the last author.

**Acknowledgments:** The fieldwork survey was supported by Nikos Bourdaniotis and Orestis Anagnopoulos of APC Advanced Planning—Consulting S.A. We would also like to thank the reviewers for their thoughtful comments and efforts toward improving our manuscript.

**Conflicts of Interest:** The authors declare no conflict of interest. The funders had no role in the design of this study; in the collection, analysis, or interpretation of the data; in the writing of the manuscript; or in the decision to publish the results.

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
