# Peer review of "Impacts of the Establishment of Biofoulants on Greek Aquaculture: Farmers’ Expert Knowledge"

_jmse, doi:10.3390/jmse11051077_

Round 1

Reviewer 1 Report

Comments and Suggestions for Authors

In the study presented in this manuscript, an interview survey has been conducted to assess mussel farmers' expert judgement of non-endemic ascidians as well as their opinions on the magnitude of the invasion's impacts, by using the LEK methodology. The goal of the study was to provide, for first time in the Eastern Mediterranean, information on species’ biofoulants in the Greek mussel farms over a long-time scale, and to describe the best management strategies that will reduce costs while preventing, controlling of biofoulants.

It is a very interesting study and this types of studies should be extended to other regions in order to propose solutions for the decision makers and the aquaculture farmers. Generally, it is well writen manuscript, though there are some suggestions, included in the attached revised manuscript, in order to further improve the way of providing the scientific results/outputs to the readers. The references are updated and the results are well presented by using the graphics.

This manuscript would be a good reference document for trying to find solutions and improvements in the management of the aquaculture farms.

Comments on the Quality of English Language

The relative comments are included in the file, though I would suggest a proof reading of the text by a native English speaker.

Author Response

We would like to thank the reviewer 1 for their constructive comments and suggestions. These were used to revise and improve the manuscript jmse-2353602 and also to clarify certain aspects that may have not been previously explained adequately. In the attach file we provide detailed responses to the comments, which were all included in the revised MS.

Reviewer 2 Report

Comments and Suggestions for Authors

In general, the manuscript requires many improvements in the writing style for the readers to understand correctly the meaning behind each sentence.

Line 48- in this section it could be useful to illustrate photographically the importance of fouling organisms to the final product appearance (after pressure washing) due to its implications in consumer rejection.

Line 76: «Farming zones have been established for shellfish farming areas at the national level, ranging from one to more than ten sample zones» The terms ‘zones’, ‘areas’ and ‘sample zones’ are quite confusing. Normally we use the expressions ‘production zone’, ‘sampling point’, regarding shellfish safety management.

Line 92: «treatment includes a variety of exposure actions» Is this only a preventive treatment? Or is the treatment of the final product mixed here? I believe ‘pressure washing’ is detrimental to mussel attachment, and is only used in the final product after harvesting.

Line 131- « Questionnaires has been given …. from entire Greek areas» Does this mean to all the producers of each area (‘all areas’)? (also an example here of improper English style)

Line 135- ‘respondents’ is better than ‘The sample’

Line 145- «The second method is more dependable but more expensive…» I believe the authors have confused the order here: the second method, ‘postal delivery’, I believe it’s cheaper than touring around the country to perform in-person interviews.

Line 251-Table 1 was not included in the manuscript for review

Line 398- what does the acronym ‘HABs’ stand for? It lacks here a brief detail of toxin production by HABs. Which types of toxins affect Greek shellfish?

Comments on the Quality of English Language

same as above

Author Response

We would like to thank the reviewer 2 for their constructive comments and suggestions. These were used to revise and improve the manuscript jmse-2353602 and also to clarify certain aspects that may have not been previously explained adequately. In the attached file, we provide detailed responses to the comments, which were all included in the revised MS.

Round 2

Reviewer 2 Report

Comments and Suggestions for Authors

The authors have improved, but I feel the manuscript was not carefully reviewed yet. Example: line 139 «the following areas areas;» the colon (:) should replace the semi-colon.

I suggest the free grammar software (https://app.grammarly.com/), it can help a lot.

Line 80 - «ranging from one to more than ten sampling point» English not ok! ‘point’ should be plural. Not clearly explained: does each area contain from one up to ten sampling points? What is the usefulness of these sampling points? Do they relate to the experimental work?

Line 101: «or are reduced the possibility» grammar not ok: ‘to reduce’ instead of ‘are reduced’

Line 152: «(2)» is not necessary.

Comments on the Quality of English Language

none

Author Response

We would like to thank the reviewer 2 for the second revision of our manuscript, which has been highly improved the presentation of the outcomes and clarified aspects that may have not been explained adequately during the first revision. In the attach file we provide detailed responses to the comments, which were all included in the revised MS.
